# Transcriptomic and Metabolomic Analysis Reveals the Potential Roles of Polyphenols and Flavonoids in Response to Sunburn Stress in Chinese Olive (*Canarium album*)

**DOI:** 10.3390/plants13172369

**Published:** 2024-08-25

**Authors:** Yu Long, Chaogui Shen, Ruilian Lai, Meihua Zhang, Qilin Tian, Xiaoxia Wei, Rujian Wu

**Affiliations:** 1Fruit Research Institute, Fujian Academy of Agricultural Sciences, Fuzhou 350013, China; llyy103699@163.com (Y.L.); shenchaogui@faas.cn (C.S.); lairuilian@faas.cn (R.L.); 2Minhou Meteorological Bureau of Fujian Province, Minhou 350100, China; fjsmhxqxj@163.com; 3Provincial Key Laboratory of Landscape Plants with Fujian and Taiwan Characteristics, Minnan Normal University, Zhangzhou 363000, China; tianql70@163.com

**Keywords:** *Canarium album*, sunburn injury, polyphenol, flavonoid, metabolome, transcriptome

## Abstract

Sunburn stress is one of the main environmental stress factors that seriously affects the fruit development and quality of Chinese olive, a tropical and subtropical fruit in south China. Therefore, the understanding of the changes in physiological, biochemical, metabolic, and gene expression in response to sunburn stress is of great significance for the industry and breeding of Chinese olive. In this study, the different stress degrees of Chinese olive fruits, including serious sunburn injury (SSI), mild sunburn injury (MSI), and ordinary (control check, CK) samples, were used to identify the physiological and biochemical changes and explore the differentially expressed genes (DEGs) and differentially accumulated metabolites (DAMs) by using transcriptomics and metabolomics. Compared with CK, the phenotypes, antioxidant capacity, and antioxidant-related enzyme activities of sunburn stress samples changed significantly. Based on DEG-based KEGG metabolic pathway analysis of transcriptomics, the polyphenol and flavonoid-related pathways, including phenylpropanoid biosynthesis, sesquiterpenoid, and triterpenoid biosynthesis, monoterpene biosynthesis, carotenoid biosynthesis, isoflavonoid biosynthesis, flavonoid biosynthesis, were enriched under sunburn stress of Chinese olive. Meanwhile, 33 differentially accumulated polyphenols and 99 differentially accumulated flavonoids were identified using metabolomics. According to the integration of transcriptome and metabolome, 15 and 8 DEGs were predicted to regulate polyphenol and flavonoid biosynthesis in Chinese olive, including 4-coumarate-CoA ligase (*4CL*), cinnamoyl-CoA reductase (*CCR*), cinnamoyl-alcohol dehydrogenase (*CAD*), chalcone synthase (*CHS*), flavanone-3-hydroxylase (*F3H*), dihydroflavonol 4-reductase (*DFR*), and anthocyanidin synthase (*ANS*). Additionally, the content of total polyphenols and flavonoids was found to be significantly increased in MSI and SSI samples compared with CK. Our research suggested that the sunburn stress probably activates the transcription of the structural genes involved in polyphenol and flavonoid biosynthesis in Chinese olive fruits to affect the antioxidant capacity and increase the accumulation of polyphenols and flavonoids, thereby responding to this abiotic stress.

## 1. Introduction

Environmental stress is a common abiotic stress that affects plant growth and development. Under natural conditions, when the environmental temperature and light intensity reach a critical threshold of plant tolerance for a long time, plants undergo irreversible damage, resulting in high-temperature stress or sunburn injury. Fruit is an important economic crop, and it is often cultivated in open fields, which are always distributed around the tree crown, and the growth and development need to be across the hot summers [1]. Long periods of high-temperature sunburn lead to shrinking or wilting of branches and leaves, causing varying degrees of burns on the surface of the fruit and even fruit drop. Protecting fruits from sunburn has become a challenge for many fruit industries.

Sunburn is a type of light damage caused by excessive heat and light (visible or ultraviolet) [2]. It has negative effects on fruit maturity, appearance, color, photosynthesis, energy metabolism, stress response and defense, cell/cell membrane/tissue structure, and so on. The injuries caused by sunburn may be exacerbated by other stress factors, such as water deprivation [3]. The increase in the levels of reactive oxygen species (ROS) is the main reason for high-temperature sunburn injury [2]. Plants have evolved effective regulatory systems, such as by increasing the biosynthesis of photoprotective compounds, heat shock protein, and nonenzymatic antioxidant metabolites, to sense and respond to high-temperature or solar radiation stress to avoid oxidative stress caused by excessive ROS accumulation [2,4]. This phenomenon conversely activates antioxidant defense mechanisms while acting in concert with antioxidant enzymes in response to excessive ROS production to improve the adaptability of plants to abiotic or biotic stresses [2,5]. For example, sunburn affects metabolic pathways, such as heat shock response, stress response and defense, energy metabolism, photosynthesis, and protein biosynthesis, during the peel color change of loquat fruit [6]. The content of chloroplasts and anthocyanin in sunburned apples decreases [7], while the content of total flavonols, dihydrochalcones, total hydroxycinnamic acid, and peroxidase activity significantly increases [8]. Meanwhile, the content of antioxidants and polyphenols in apples increases after long-term strong light irradiation to adapt to the strong light environment [9]. With the increase in sunburn injury, photosynthetic pigments are destroyed, and polyphenol content is significantly increased to protect grapes from sunburn injury [10]. Sunburn injury leads to a decrease in the chlorophyll content, an increase in the flavonoid and carotenoid contents, and a higher ability to scavenge free radicals in the peel of satsuma mandarin fruits [11]. The content of total phenols and total flavonoids increases in sunburned pomegranate fruits, enhancing the total antioxidant capacity and thereby reducing and repairing membrane damage [12]. The sunburn injury promotes the accumulation of polyphenols and enhances the activity of antioxidant enzymes to increase the anti-oxidation ability of peels in pear [13].

The response of plants to sunburn stress depends on multiple signal transduction, metabolite biosynthesis, and gene transcription [14,15,16]. Metabolomics, based on high-throughput data, effectively identifies the dynamic composition of endogenous metabolites in organisms. Transcriptomics, on the other hand, can effectively monitor all gene transcripts in biological samples. Both of them are widely used in the study of plant stress response mechanisms. The transcriptomic and metabolomic studies revealed that sunburn caused significant changes in the content of 129 metabolites and the transcription of 447 genes in pomegranate. *NAC5*, *MYB93*, and *MYB111* transcription factors may be involved in phenylpropanoid and flavonoid biosynthesis in response to sunburn stress by regulating chalcone isomerase (*CHI*), flavonoid 3′, 5′-hydroxylase (*F3’5’H*), and chalcone synthase (*CHS*) [12]. Flavonoid biosynthesis and heat shock protein–heat shock factor pathway of *Sorbus pohuashanensis* played an important role in resisting sunburn stress [17]. The pathway involved in ROS clearance in apples was potentially crucial in response to sunburn stress [18]. These studies revealed that different plants showed different response modes to sunburn stress.

Chinese olive (*Canarium album* Raeusch.) is a traditional Chinese medicine and characteristic fruit and is widely distributed in southern China [19]. As a typical tropical and subtropical crop, the Chinese olive plant has a strong high-temperature tolerance. However, its fruits are often distributed outside the tree crown. During the fruit development and expansion period from July to August each year, they are prone to extremely high temperatures and strong sunlight, leading to severe diseases in fruits and physiological fruit drop. Hence, sunburn stress has become one of the essential environmental factors restricting the Chinese olive industry. This experiment identified the physiological and biochemical changes in Chinese olive fruits after sunburn stress and explored differentially accumulated metabolites (DAMs) and differentially expressed genes (DEGs) during sunburn stress occurrence through the integration of metabolomic and transcriptomic analyses. A metabolic regulatory network for the response of Chinese olive plants to sunburn stress was preliminarily constructed, laying a theoretical foundation for the study of genetic mechanisms of defense of Chinese olive plants and the selection of high-resistance varieties.

## 2. Results

### 2.1. Changes in Phenotypic and Antioxidant Capacity

The changes in the phenotype, color, and firmness of Chinese olive fruits under sunburn stress are shown in Figure 1. The results indicated that the sunburn caused varying degrees of burned patches on the surface of Chinese olive fruits (Figure 1A). In terms of fruit color (Figure 1B), the *a** value of the mild sunburn injury (MSI) sample significantly increased, whereas the *h** value significantly decreased. However, the *a** value of the serious sunburn injury (SSI) sample significantly increased, whereas the *L**, *b**, *c**, and *h** values all significantly decreased compared with those of the control check (CK) sample. Significant differences in chromatic aberration parameters were found between MSI and SSI samples, except for the *a** value. The fruit firmness of the MSI sample did not show significant change (Figure 1C), while that of the SSI sample significantly improved compared with that of the CK sample. The fruit firmness of the SSI sample also considerably increased compared with that of the MSI sample. In conclusion, MSI led to changes in the color of the fruit, whereas SSI affected both the color and firmness of the fruit.

Further analysis found that the total antioxidant capacity (T-AOC) changed under sunburn stress. Compared with the CK sample, T-AOC was significantly upregulated in the MSI sample and significantly decreased in the SSI sample (Figure 1D). The hydrogen peroxide (H_2_O_2_) content significantly decreased in the MSI sample; however, there was no change in the SSI sample (Figure 1E). Moreover, the contents of malondialdehyde (MDA) and proline both significantly increased in MSI and SSI samples compared with the CK sample (Figure 1F,G). Compared with the CK sample, the activities of several antioxidant-related enzymes were significantly changed (Figure 1H); the catalase (CAT), superoxide dismutase (SOD), and ascorbate peroxidase (APX) activities increased in the MSI sample and then decreased in the SSI sample, while that of polyphenol oxidase (PPO) and peroxidase (POD) continuously increased.

### 2.2. Transcriptomic Analysis

#### 2.2.1. Transcriptome Data Evaluation and DEGs Identification

A total of 72.16 GB of clean data were obtained from three biological duplicate samples of T-CK, T-MSI, and T-SSI using Illumina Hiseq 2000 (Biomarker Technologies, Beijing, China), and 43,160 unigenes were annotated. Meanwhile, the clean data range for each sample was from 7.36 to 8.88 GB, with the GC base content ranging from 43.74% to 44.06% and Q30 exceeding 96%. The data quality and reliability of the constructed library were found to be high and hence used for subsequent analysis.

A total of 3259 DEGs (Appendix A) were selected based on the criteria of the corrected *p* value < 0.01 and |log2 ^fold change (FC)^| ≥ 1.0. Among these, 2110, 2340, and 65 DEGs were identified between the comparison groups of T-CK vs. T-MSI, T-CK vs. T-SSI, and T-MSI vs. T-SSI, respectively. Among these, 1216, 34, and 8 DEGs were found to have differences between T-CK vs. T-MSI and T-CK vs. T-SSI, between T-CK vs. T-SSI and T-MSI vs. T-SSI, and between T-CK vs. T-MSI and T-MSI vs. T-SSI, respectively (Appendix A). The cluster analysis results indicated that the variation patterns of DEGs in MSI and SSI samples were similar (Appendix A), indicating that most DEGs had consistent responses to mild and serious sunburn injuries.

#### 2.2.2. GO Pathway Enrichment Analysis Based on DEGs

Based on Gene Ontology (GO) enrichment analysis, the DEGs between samples were divided into cellular component (CC), molecular function (MF), and biological process (BP). Among these, 1635 DEGs between T-CK and T-MSI were enriched into 18 BPs, 15 CCs, and 12 MFs related to GO pathways (Appendix A). Also, 1805 DEGs between T-CK and T-SSI were annotated into 18 BPs, 17 CCs, and 13 MFs related to GO pathways (Appendix A). Additionally, DGEs between T-MSI and T-SSI samples were enriched into 24 GO pathways (12 BPs, 8 CCs, and 4 MFs) (Appendix A). Among these, 44, 24, and 24 GO pathways co-enriched between T-CK vs. T-MSI and T-CK vs. T-SSI, T-CK vs. T-MSI and T-MSI vs. T-SSI, and T-CK vs. T-SSI and T-MSI vs. T-SSI, respectively. These results indicated that the GO pathways enriched in the SSI sample were similar to those in the MSI sample. Further analysis found that there were 20 unique co-enriched GO pathways between T-CK vs. T-MSI and T-CK vs. T-SSI without in T-MSI vs. T-SSI, including BPs such as reproduction, reproductive process, growth, rhythmic process, and detoxification, CCs such as cell junction, membrane-enclosed lumen, other organism, other organism part, extracellular region part, symplast, and supramolecular complex, MFs such as transcription factor activity, protein binding, signal transducer activity, structural molecule activity, electron carrier activity, antioxidant activity, nutrient reservoir activity, molecular transducer activity, molecular function regulator. Among these, it was worth noting that some DEGs were enriched in the antioxidant activity GO pathway, which was similar to the determination of physiological indicators related to antioxidant activity. Moreover, the main GO pathways enriched in T-CK vs. T-MSI, T-CK vs. T-SSI, and T-MSI vs. T-SSI (including more than 10% of DEGs) were completely consistent, mainly including BPs such as metabolic processes, cellular processes, single organism processes, biological regulation, and response to stimuli, CCs such as cell, cell part, membrane, organelle, and membrane part, as well as the binding and catalytic activities of MFs. This indicated that these 12 GO pathways played potentially important roles in the whole response process of Chinese olive fruits to sunburn stress.

#### 2.2.3. Enrichment Analysis of KEGG Metabolic Pathways Based on DEGs

The DEGs in comparisons of T-CK vs. T-MSI, T-CK vs. T-SSI, and T-MSI vs. T-SSI were enriched into 122, 126, and 31 Kyoto Encyclopedia of Genes and Genomes (KEGG) metabolic pathways, respectively. Among these, 120, 30, and 31 KEGG metabolic pathways co-enriched between T-CK vs. T-MSI and T-CK vs. T-SSI, T-CK vs. T-MSI and T-MSI vs. T-SSI, and T-CK vs. T-SSI and T-MSI vs. T-SSI, respectively. These results indicated that the KEGG metabolic pathway enriched after mild and serious sunburn stresses in Chinese olive fruits was similar. Further analysis of the top 20 enriched metabolic pathways revealed that T-CK vs. T-MSI and T-CK vs. T-SSI had 14 pathways in common (Figure 2A,B), including phenylpropanoid biosynthesis, plant hormone signal transduction, MAPK signaling pathway–plant, sesquiterpenoid and triterpenoid biosynthesis, monoterpenoid biosynthesis, tryptophan metabolism, cutin, suberine, and wax biosynthesis, galactose metabolism, isoflavonoid biosynthesis, flavonoid biosynthesis, pentose and glucuronate interconversions, starch and sucrose metabolism, carotenoid biosynthesis, and brassinosteroid biosynthesis. Moreover, the pathways of galactose metabolism and isoflavonoid biosynthesis were also found in the top 20 enriched KEGG metabolic pathways in comparison of T-MSI vs. T-SSI (Figure 2C). Interestingly, several metabolic pathways related to polyphenols and flavonoids, such as phenylpropanoid biosynthesis, sesquiterpenoid and triterpenoid biosynthesis, monoterpene biosynthesis, isoflavonoid biosynthesis, flavonoid biosynthesis, carotenoid biosynthesis, were found to be enriched under sunburn stress of Chinese olive, indicating that polyphenols and flavonoids might be potentially regulatory metabolites for sunburn injury of Chinese olive fruits.

### 2.3. Metabolomic Analysis

#### 2.3.1. Metabolome Data Assessment and Metabolite Identification

Based on widely targeted metabolomics, 3380 metabolites (Appendix A) were identified from M-CK, M-MSI, and M-SSI samples. Principal component analysis (PCA) showed that the biological duplicate samples within the group could aggregate well, indicating good-quality metabolomic data. Further, the first and second principal components reflected 36.89% and 19.03% of the data features, respectively (Appendix A). The correlation analysis revealed that the square of the Spearman correlation coefficient r between biological repeated samples within the group was greater than 0.9, further indicating good biological repeatability (Appendix A). The positive correlation between M-MSI and M-SSI was strong, indicating that the metabolic components after mild and serious sunburn stresses were similar compared with those for CK samples. These results indicated that metabolome data have high reliability and can be used for further analysis.

A total of 800 DAMs (Appendix A) were identified using variable importance in projection (VIP) ≥ 1.0, |log2^FC^| ≥ 1.0, and *p* value < 0.05 as the screening criteria. DAM cluster analysis revealed that the metabolite composition of M-MSI and M-SSI was relatively similar (Figure 3A), which was consistent with the transcriptome sequencing results. Among these, 532, 501, and 180 DAMs were identified in comparisons of M-CK vs. M-MSI, M-CK vs. M-SSI, and M-MSI vs. M-SSI, respectively. Further analysis found that 282, 81, and 66 DAMs shared differences both in M-CK vs. M-MSI and M-CK vs. M-SSI, M-CK vs. M-MSI and M-MSI vs. M-SSI, and M-CK vs. M-SSI and M-MSI vs. M-SSI, while 16 DAMs were differentially accumulated in all comparisons (Figure 3B).

#### 2.3.2. KEGG Pathway Analysis of DAMs

Based on DAMs, there were 97, 86, and 60 KEGG enrichment pathways in M-CK vs. M-MSI, M-CK vs. M-SSI, and M-MSI vs. M-SSI, respectively. According to the order of *p* value of the KEGG pathway (top 20), it was worth noting that some flavonoid biosynthesis-related pathways were significantly enriched or enriched in M-CK vs. M-MSI, M-CK vs. M-SSI, and M-MSI vs. M-SSI, such as isoflavonoid biosynthesis, flavone and flavonol biosynthesis, flavonoid biosynthesis, and anthocyanin biosynthesis (Appendix A). These results were consistent with the KEGG enrichment analysis of DEGs, which showed that the flavonoid-related metabolic pathways might take part in response to sunburn stress.

### 2.4. Integration of Transcriptome and Metabolome

#### 2.4.1. Changes of Polyphenol Related Metabolites and Genes

The transcriptomic analysis results revealed that polyphenols and flavonoids were important metabolites of Chinese olive fruits that potentially responded to sunburn stress. Based on metabolomic data, a total of 33 differentially accumulated polyphenols (DAPs) were found in sunburn stress samples of Chinese olive, including 31 up-accumulated components, and involving cinnamaldehydes, cinnamic acids, coumarins, isocoumarins, diarylheptanoids, phenylpropanoic acids, tannins, and their derivatives (Appendix A). Further, 9 DAPs in metabolic pathway of phenylpropanoid biosynthesis were found (Figure 4), of which one DAP was down-accumulated (4-hydroxycinnamic alcohol 4-*D*-glucoside) and eight DAPs were up-accumulated, including cinnamic acid, anethole, caffeoyl quinic acid, caffeoyl-CoA, caffeoyl aldehyde, *p*-coumaraldehyde, feruloyl-CoA, and coniferin. Meanwhile, 15 DEGs of this metabolic pathway were identified from transcriptomic data, including seven of 4-coumarate-CoA ligases (*4CLs*), three of caffeic acid 3-*O*-methyltransferases (*COMTs*), two of cinnamoyl-alcohol dehydrogenase (*CADs*), one of cinnamoyl-CoA reductase (*CCR*), phenylalanine ammonia lyase (*PAL*), and coniferyl-alcohol glucosyltransferase (*CAGT*). It was worth noting that the expression levels of several genes, such as *4CL* (BMK_Unigene_064108, BMK_Unigene_102104), *CCR* (BMK_Unigene_055431), and *CAD* (BMK_Unigene_068171), were changed more than two times in different sunburn stress samples, indicating that they might play important roles in response to this stress of Chinese olive.

#### 2.4.2. Changes of Flavonoid-Related Metabolites and Genes

A total of 99 differentially accumulated flavonoids (DAFs) in different sunburn stress samples were identified by metabolomics, including 88 up-accumulated components, which were involved in flavones, flavonols, isoflavonoids, isoflavans, chalcones, dihydrochalcones, flavans, flavanols and leucoanthocyanidins, flavanones, and anthocyanidins (Appendix A). Among them, 16 DAFs were differentially accumulated in the metabolic pathway of flavonoid biosynthesis (Figure 5), of which 6 DAFs were down-accumulated in comparison to M-MSI vs. M-SSI, and 16 DAFs were significantly up-accumulated in comparison of M-CK vs. M-MSI or M-CK vs. M-SSI. Based on transcriptomic data, a total of eight structural genes of this pathway were identified as DEGs, including two of *CHSs* and one of *CHI*, flavanone 7-*O*-glucoside 2″-*O*-beta-L-rhamnosyltransferase (*FGR*), flavanone-3-hydroxylase (*F3H*), dihydroflavonol 4-reductase (*DFR*), anthocyanidin synthase (*ANS*), and flavonol synthase (*FLS*). Notably, the expression levels of several genes, such as *CHSs* (BMK_Unigene_068778 and BMK_Unigene_104115), *F3H* (BMK_Unigene_099272), *DFR* (BMK_Unigene_072482), and *ANS* (BMK_Unigene_068890), significantly increased during sunburn stress, indicating that they potentially regulated the response of Chinese olive to this stress by mediating flavonoid biosynthesis.

#### 2.4.3. Changes of Transcription Factors Related to Regulating Phenylpropanoid and Flavonoid Biosynthesis

The accumulation of polyphenols and flavonoids in plants is not only regulated by structural genes directly encoding enzymes related to phenylpropanoid and flavonoid synthesis but is also affected by transcription factors, including *MYB* and *bHLH*. Related studies indicated that *MYB* and *bHLH* can act as transcriptional activators or inhibitors, individually or jointly regulating the expression levels of structural genes related to phenylpropanoid and flavonoid biosynthesis pathways, thereby increasing or inhibiting the accumulation of polyphenols and flavonoids in plants, and mediating the regulation of plant tolerance to stress [21,22,23]. Based on transcriptomic data, a total of 32 differentially expressed *MYBs* after sunburn stress were identified (Appendix A), of which 16 *MYBs* had higher expression levels in CK, MSI, and SSI samples, while the other 16 *MYBs* had lower expression abundances. What was noteworthiness was that the BMK_Unigene_013290 continuously down-expressed, while the BMK_Unigene_063026, BMK_Unigene_063155, and BMK_Unigene_064813 continuously up-expressed, which were potential negative and positive regulation of *MYB* transcription factors. Moreover, the expressions of 19 *bHLHs* were found to be significantly changed during this process. The overall expression abundances of seven *bHLHs* were higher, while the expression abundance of 12 *bHLHs* was lower. Especially, the expression levels of three *bHLHs* (BMK_Unigene_023520, BMK_Unigene_022454, BMK_Unigene_024971) were significantly decreased after sunburn stress, which potentially negatively regulated the response of Chinese olive to this stress.

#### 2.4.4. Changes of Antioxidant Capacity-Related Genes

DEG-based GO pathway enriched analysis found that the antioxidant activity pathway was one of the specific co-enriched GO pathways in T-CK vs. T-MSI and T-CK vs. T-SSI. Further analysis indicated that a total of 13 differentially expressed genes related to antioxidant capacity were identified (Appendix A), including one *CAT*, three *APX*, and nine *POD* genes. Among these, one *CAT* gene (BMK_Unigene_025872) was significantly down-expressed in both T-CK vs. T-MSI and T-CK vs. T-SSI. Three *APXs* were significantly up-expressed, and the fold change of BMK_Unigene_096922 was the highest in both T-CK vs. T-MSI and T-CK vs. T-SSI. Among the nine *POD* genes, there were four identical genes in both T-CK vs. T-MSI and T-CK vs. T-SSI, including three significantly up-expressed genes and one significantly down-expressed gene. It was worth noting that BMK_Unigene_099223 and BMK_Unigene_018627 were the two *POD* genes with the highest and lowest fold change, respectively. These genes might respond to sunburn stress by regulating the activity of corresponding enzymes in Chinese olive, including one *CAT* (BMK_Unigene_025872), one *APX* (BMK_Unigene_096922), and two *PODs* (BMK_Unigene_099223, BMK_Unigene_018627).

### 2.5. Changes in the Content of Total Polyphenols and Total Flavonoids

Metabolomic and transcriptomic data revealed that polyphenols and flavonoids potentially responded to sunburn stress in Chinese olive. Compared with the CK sample, the content of total polyphenols and total flavonoids (Figure 6) significantly increased by 1.65 and 2.00 times in the MSI sample and by 1.57 and 1.45 times in the SSI sample, respectively. And no significant difference in the content of total polyphenols was found between the MSI and SSI samples, while the content of total flavonoids significantly decreased in the SSI sample than those in the MSI sample. The results suggested that sunburn stress resulted in the accumulation of polyphenols and flavonoids in Chinese olive.

## 3. Discussion

### 3.1. Serious Sunburn Led to Significant Changes in Phenotype of Chinese Olive

Fruit color and firmness are essential phenotypic indicators for evaluating fruits. Studies on citrus [11], apple [7,8], and grape [24] have shown that sunburn affects the accumulation of pigments in the skin or flesh of fruit. In this study, the colors of Chinese olive after different types of sunburn stress significantly changed. These results were consistent with previous findings. Kim et al. [11] found that the firmness of high-temperature sunburn of satsuma mandarin fruits significantly increased, which might be related to the shrinkage, roughness, and cracking of the fruit surface under anatomical observation, ultimately leading to necrosis and browning caused by pathogen invasion. Torres et al. [25] revealed that photooxidative stress increased the firmness of apples, which was involved in the induction of lignin accumulation of stressed fruit, with the involvement of stress phytohormones such as ethylene. The changes in fruit color, phenotypic browning, or cell necrosis and hardening caused by high-temperature sunburn were closely related to oxidative stress caused by excessive ROS production [9,26]. In our present study, the mild sunburn stress did not cause obvious changes in the firmness of Chinese olive, while the serious sunburn injury significantly increased the firmness of the fruits. These phenomena might be related to the necrosis and pathological changes caused by serious sunburn.

### 3.2. Sunburn Stress Up-Accumulated the Polyphenol and Flavonoid in Chinese Olive to Involve in the Response to This Abiotic Stress

Sunburn is a common abiotic stress during growth and development of plants. Continuous high temperature and strong light cause metabolic disorders related to heat stress, leading to excessive formation of ROS and disruption of plant cell homeostasis [27]. Plants biosynthesize photoprotective compounds, heat shock proteins, and nonenzymatic antioxidants (polyphenols and flavonoids), enhance antioxidant enzyme activity, and improve plant oxidative stress capacity to alleviate the oxidative damage caused by excessive ROS to plants [2,5,28]. Polyphenols and flavonoids are important secondary metabolites in plants, which have been proven to have good antioxidant activity and are essential active substances to improve abiotic stress tolerance and adaptability [29,30,31,32]. Previous studies showed that sunburn promoted the accumulation of polyphenol and flavonoid compounds in pomegranate [12], apple [8,9,33], grape [10], satsuma mandarin [11], sea buckthorn [34], and strawberry [35]. Moreover, the polyphenol and flavonoid components have been proven to regulate plant antioxidant capacity. For example, the exogenous proanthocyanidin treatment inhibited the senescence of harvested banana fruit by enhancing antioxidant response and in vivo proanthocyanidin content [36]. Exogenous chlorogenic acid alleviated oxidative stress in apple leaves by enhancing antioxidant capacity [37]. Exogenous cinnamic acid regulated antioxidant enzyme activity and reduced lipid peroxidation in drought-stressed cucumber leaves [38]. In this study, the content of total polyphenols and total flavonoids significantly increased, and most of the DAPs and DAFs up-accumulated after sunburn stress in Chinese olive, suggesting that the polyphenol and flavonoids potentially regulated the response to this stress in Chinese olive. However, compared to the mild sunburn stress sample, the content of polyphenols and flavonoids in the serious sunburn stress sample decreased to a certain extent. Generally, when plants are exposed to stress, as the stress intensifies, the first-level response may not be sufficient, and plants may activate the second-level response [39]. It could be predicted that, after mild sunburn injury, the cells responded to the stress immediately and biosynthesized and accumulated the stress resistance-related metabolites, such as polyphenols, flavonoids, and antioxidant enzymes. However, the injury exceeded the threshold that the cells could withstand, resulting in significant cell or tissue necrosis.

### 3.3. The Potential Regulatory Network of Polyphenol and Flavonoid in Response to Sunburn Stress in Chinese Olive

Previous reports have shown that in pomegranates, sunburn stress led to significant changes in the expression levels of *CHI*, *F3*′*5*′*H,* and *CHS* genes [12]. Transcriptome DEG-based KEGG analysis revealed that phenylpropanoid biosynthesis was involved in the response of Chinese olive to sunburn stress. Moreover, 15 genes in this pathway were significantly differentially expressed during stress occurrence, including *4CL*, *COMT*, *PAL*, *CCR*, *CAD*, *CAGT*, and so forth. Among these, two *4CLs* (BMK_Unigene_064108, BMK_Unigene_102104), one *CCR* (BMK_Unigene_055431), and one *CAD* (BMK_Unigene_068171) were key genes potentially regulating polyphenol biosynthesis. Moreover, eight potential regulatory genes of flavonoid biosynthesis were identified, including *CHS*, *CHI*, *FGR*, *F3H*, *DFR*, *ANS*, and *FLS*, and the *CHSs* (BMK_Unigene_068778, BMK_Unigene_104115), *F3H* (BMK_Unigene_099272), *DFR* (BMK_Unigene_072482), and *ANS* (BMK_Unigene_068890) were screened as candidate genes for regulating flavonoid accumulation and sunburn stress response.

H_2_O_2_ is important for ROS; biotic and abiotic stresses have been revealed to induce the production and accumulation of H_2_O_2_ in plant cells [40]. MDA is the main product of membrane lipid peroxidation, and its content can be used to assess cell membrane damage and plant tolerance [41]. The accumulation of proline also reflects the injury of plants under stress [42], and the accumulated proline could function to clear ·OH [43]. Studies on citrus [44], apple [9], and pomegranate [12] have shown that sunburn stress led to a significant increase in O_2_− and MDA content. As sunburn damage occurred, they up-accumulated and destroyed the cell membrane system. This study found that after sunburn stress, the MDA accumulation significantly increased, indicating damage to the cell membrane system in Chinese olive. During this process, the content of T-AOC and proline significantly increased, while the content of H_2_O_2_ was decreased, indicating that the accumulation of proline helped to eliminate excessive ROS and played an important role in antioxidants. Meanwhile, the antioxidant-related enzymes were activated, and multiple antioxidant enzymes were jointly involved in regulating the process [9]. In this study, MSI induced significant upregulation of activities of CAT, SOD, APX, POD, and PPO, indicating these enzymes are involved in sunburn stress in Chinese olive. Moreover, the expression levels of antioxidant regulatory genes have also been significantly increased, including *CAT*, *APX,* and *POD*. It could be seen that after sunburn stress, the antioxidant capacity in Chinese olive was enhanced by upregulating the regulatory gene expression to increase the activities of antioxidant-related enzymes.

In summary, flavonoids and polyphenols are important secondary metabolites in Chinese olive. Sunburn stress might activate the transcription of structural genes, such as *4CL*, *CCR*, *CAD*, *CHS*, *F3H*, *DFR*, and *ANS*, to mediate the biosynthesis and accumulation of polyphenols and flavonoids, thereby responding to sunburn stress (Figure 7).

## 4. Materials and Methods

### 4.1. Materials

The samples used in this study were the fruits of the *Canarium album* cv. *lingfeng*, a fresh edible cultivar selected by the Fruit Research Institute of Fujian Academy of Agricultural Sciences, and the samples were collected on July 29, 2022, from the experimental base located in Zhenjing Farm, Baisha, Minhou (26.21° N, 118.99° E, A119.2 m). Data from the orchard meteorological observation station revealed that the daily precipitation of the orchard on the sampling day was 0.0 mm. The daily average temperature was recorded at 31.4 °C, with a daily maximum of 36.9 °C and a daily minimum of 27.1 °C. The sunshine duration on that day was 9.8 h. The daily average temperature of the last 20 days before sampling was 31.6 °C, with a maximum temperature of 41.3 °C (average 38.0 °C). The daily maximum temperature of 18 days exceeded 35.0 °C, and the longest sunshine hours reached 12.7 h, of which 19 days had almost no precipitation (Appendix A). The sunburn injury in Chinese olive fruits caused by sustained abnormal high temperature and strong sunlight is shown in Figure 1A. Normal fruit without sunburn injury was used as a control check (CK) sample, fruit with symptom and sunburn injury area of less than one-fourth of the whole fruit area were used as the mild sunburn injury (MSI) sample, and fruit with sunburn injury area of greater than one-fourth of the whole fruit area or those beginning to experience fruit tissue necrosis were used as the serious sunburn injury (SSI) sample. Five trees (numbered Ca1–Ca5) were randomly selected from the orchard; the grafted age of each tree was more than 9 years, and 10 fruits with different degrees of sunburn injury were collected from the same sunny side of each tree. The collected samples were taken to the laboratory, and the flesh was quickly cut. The samples were wrapped in tin foil, quick-frozen with liquid nitrogen, and placed in an ultra-low temperature freezer at −80 °C for later use.

### 4.2. Methods

#### 4.2.1. Detection of Physiological and Biochemical Indicators

The color and firmness of the pericarp were determined using an NH310 chroma meter (3nh Technology Co., Ltd., Shenzhen, China) and a TMS-Pilot texture analyzer (Food Technology Corporation, Sterling, VA, USA). The content of total polyphenols was determined by a modified Folin–Ciocalteu method with gallic acid as a control [45]. Rutin was used as a control, and spectrophotometry was used to determine total flavonoids [46]. The contents of proline, T-AOC, H_2_O_2_, MDA, SOD, POD, APX, CAT, and PPO of samples were detected using the method described by ninhydrin colorimetric method [47], ferric reducing ability of plasma method [48], ammonium molybdate method [49], thiobarbituric acid method [50], nitro-blue tetrazolium method [50], guaiacol method [50], ascorbic acid colorimetric method [51], pyrocatechol method [47]. Each sample was measured for five biological replicates.

#### 4.2.2. Transcriptome Sequencing

The CK, MSI, and SSI samples used for transcriptome sequencing were named T-CK, T-MSI, and T-SSI for library construction and sequencing, respectively, and three biological replicates were set up. First, the E.Z.N.A Plant RNA Kit (OMEGA Biotek Inc., Norcross, GA, USA) was used to extract the total RNA of the samples, agarose gel electrophoresis and TU-1810 spectrophotometer (Puxi General Instrument Co., Ltd., Beijing, China) were used to detect its integrity and purity, and an Agilent 2100 biological analyzer (Agilent Technologies, Santa Clara, CA, USA) was used to evaluate the RNA quality. After passing the inspection, 1 μg total RNA was taken, and sequencing libraries were constructed using the NEBNext Ultra RNA Library Prep Kit for Illumina (New England Biolabs, Ipswich, MAS, USA). Briefly, mRNA was purified from total RNA using poly-T oligo-attached magnetic beads. Fragmentation was carried out using divalent cations at elevated temperatures in NEBNext first-strand synthesis reaction buffer (5×). The first-strand cDNA was synthesized using a random hexamer primer and M-MuLV reverse transcriptase. The second-strand cDNA synthesis was subsequently performed using DNA polymerase I and RNase H. The library fragments were purified with an AMPure XP system (Beckman Coulter, Brea, CA, USA). Then, 3 μL USER enzyme (New England Biolabs, Ipswich, MAS, USA) was used with cDNA at 37 °C for 15 min and then at 95 °C for 5 min for polymerase chain reaction to construct the library. The library quality was assessed using the Agilent Bioanalyzer 2100 (Agilent Technologies, Santa Clara, CA, USA)and sequenced using the Illumina Hiseq 2000 platform (Biomarker Technologies, Beijing, China).

The *Arabidopsis thaliana* genome was the reference for transcriptome assembly. The transcripts were annotated based on NCBI non-redundant protein sequences (NR), Protein family (Pfam), Clusters of Orthologous Groups of proteins (KOG/COG/eggNOG), Swiss-Prot, Kyoto Encyclopedia of Genes and Genomes (KEGG), and Gene Ontology (GO). Differential expression analysis of two groups was performed using the DESeq R package (v 1.30.1). The expression abundance of the two groups was expressed by fragments per kilobase of transcript per million mapped reads. The corrected *p* value < 0.01 and fold change (FC) ≥ 2.0 or ≤0.5 (i.e., |log2^FC^| ≥ 1.0) were used as the criteria to screen DEGs, and GO and KEGG pathway enrichment analyses were performed based on DEGs.

#### 4.2.3. Metabolomic Assay

The CK, MSI, and SSI samples used for metabolomic detection were named M-CK, M-MSI, and M-SSI, respectively. The quality control (QC) samples were mixed in equal amounts, and three biological replicates were set up for nontargeted metabolomic detection. Then, 50 mg of the sample was used, and 1 mL of extraction solution containing internal standard (methanol–acetonitrile–water = 2:2:1, the internal standard concentration of 20 mg·L^−1^) was added for extraction. A liquid chromatography–mass spectrometry system consisting of Waters UPLC Acquisition I-Class PLUS (Waters, Milford, MAS, USA) and a high-resolution mass spectrometer (UPLC Xevo G2-XS QTOF, Waters, Milford, MAS, USA) was used for metabolomic detection. The used column was Acquity UPLC HSS T3 (1.8 µm 2.1 mm × 100 mm Waters, Milford, MAS, USA). Both in positive and negative ion modes, mobile phase A and mobile phase B were composed of 0.1% formic acid aqueous solution and 0.1% formic acid acetonitrile, respectively, with injection volumes of 1 μL. Liquid chromatography mobile-phase conditions: flow rate, 400 μL·min^−1^; 0–0.25 min, 98% A, 2% B; 0.25–10.00 min, 98% A, 2% B; 10.00–13.00 min, 2% A, 98% B; 13.00–13.10 min, 98% A, 2% B; and 13.10–15.00 min, 98% A, 2% B. In each data acquisition cycle, dual-channel data acquisition was performed on both low- and high-collision energies. The scanning frequency was 0.2 s, and the electron spray ionization (ESI) source parameters were set according to the method described by Wang et al. [52].

The MassLynx (v 4.2) was used to import the collected raw data into the Progenesis QI (v 2.0) software for peak extraction, peak alignment, and so on, based on the Progenesis QI software online METLIN database, public database, and Biomark’s database for metabolite identification. Meanwhile, the theoretical fragment was identified, the precursor ion mass deviation was 100 ppm, and the fragment ion mass deviation was within 50 ppm [52]. The metabolites were quantitatively detected. KEGG, Human Metabolome Database (HMDB), and Lipid Metabolites and Pathways Strategy (LIPID MAPS) public databases were used for metabolite function and classification annotation. After normalizing the quantitative values of metabolites, principal component analysis (PCA) and Spearman’s correlation analysis were used to judge the repeatability of the samples within the group and the quantity control samples. Orthogonal projections to latent structures–discriminant analysis was performed. FC ≥ 2.0 or ≤ 0.5 (i.e., |log2^FC^| ≥ 1.0), variable importance in projection (VIP) ≥ 1.0, and significance *p* value < 0.05 were set to screen DAMs.

#### 4.2.4. Statistical Analysis

Analyses of physiological and biochemical data were conducted using Microsoft Excel (v 2013) and IBM SPSS Statistics (v 19.0) software. Results were expressed as mean ± SD. *p* values of ≤0.01 and 0.05 were considered a statistically significant difference between treatments by the LSD method, which were represented in upper- and lowercase letters, respectively.

## 5. Conclusions

Chinese olive fruits underwent significant changes in color and increases in fruit firmness, antioxidant capacity, and antioxidant enzyme activities after sunburn injury. Based on transcriptomic identification, a total of 3259 DEGs were obtained, including 15 phenylpropanoid and 8 flavonoid biosynthesis regulatory structural genes and 7 key transcription factors. Based on metabolomic identification, a total of 800 DAMs were obtained, including 33 DAPs and 99 DAFs. DEG- and DAM-based KEGG revealed that sunburn stress significantly activated the polyphenol and flavonoid-related biosynthetic pathways in Chinese olive fruits, and the content of total polyphenols and total flavonoids significantly increased after the stress. These results indicated that sunburn stress might activate the transcription of structural genes, such as *4CL*, *CCR*, *CAD*, *CHS*, *F3H*, *DFR*, and *ANS*, so as to increase the accumulation of the polyphenols and flavonoids to enhance the tolerance of Chinese olive fruits to sunburn stress.

## Figures and Tables

**Figure 1 plants-13-02369-f001:**
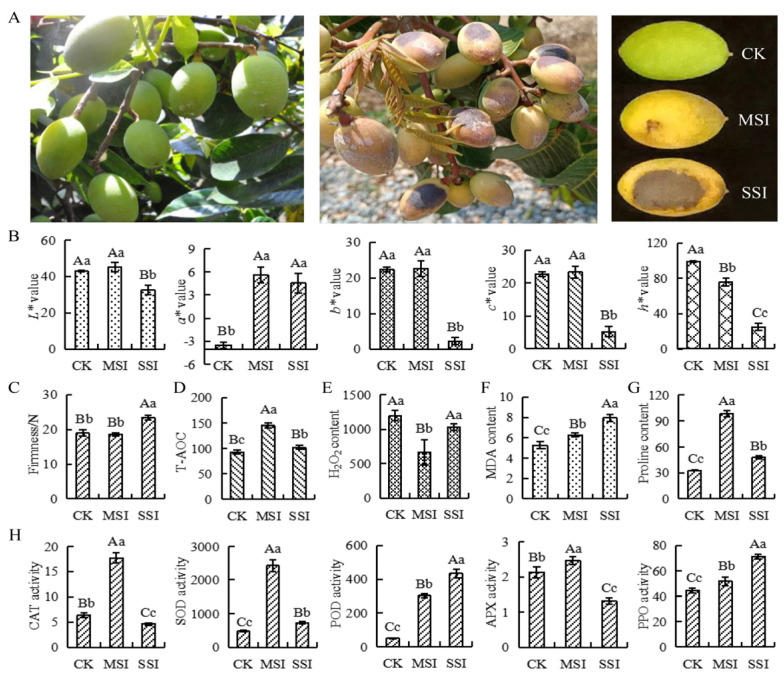
The changes of phenotypes, antioxidant capacity, and antioxidant-related enzyme activities in response to sunburn stress of Chinese olive. (**A**) The phenotype changes of samples. (**B**) The color changes of samples are represented by chromatic aberration *L**, *a**, *b**, *c**, and *h** values. (**C**) The firmness changes of samples. (**D**) The T-AOC changes of samples (μmol·mL^−1^ FW). (**E**) The content of H_2_O_2_ in samples (μmol·g^−1^ FW). (**F**) The content of MDA in samples (nmol·g^−1^ FW). (**G**) The content of proline in samples (μg·g^−1^ FW). (**H**) The activity changes of CAT, SOD, POD, APX, and PPO of samples (U·g^−1^ FW). CK, MSI, and SSI indicate the samples of control check, mild sunburn injury, and serious sunburn injury, respectively. The upper- and lowercase letters on the curves represent significant differences at *p* value < 0.01 and <0.05 levels.

**Figure 2 plants-13-02369-f002:**
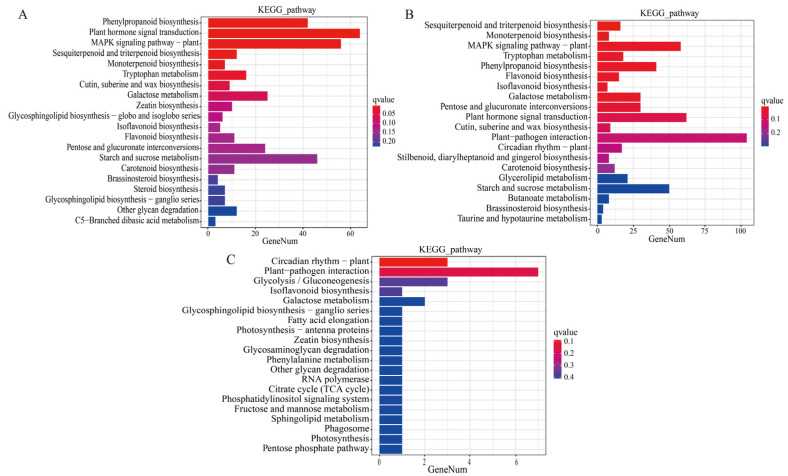
DEG-based enrichment analysis of KEGG metabolic pathway. (**A**–**C**) indicate the comparisons of T-CK vs. T-MSI, T-CK vs. T-SSI, and T-MSI vs. T-SSI. The red and blue colors indicate high and low q values, respectively.

**Figure 3 plants-13-02369-f003:**
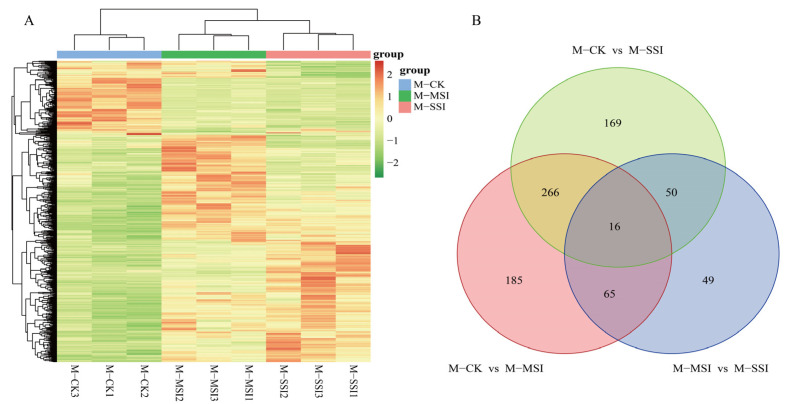
DAMs identification. (**A**,**B**) indicate the cluster analysis and Venn diagram. M-CK, M-MSI, and M-SSI indicate the metabolomes of control check, mild sunburn injury, and serious sunburn injury, respectively.

**Figure 4 plants-13-02369-f004:**
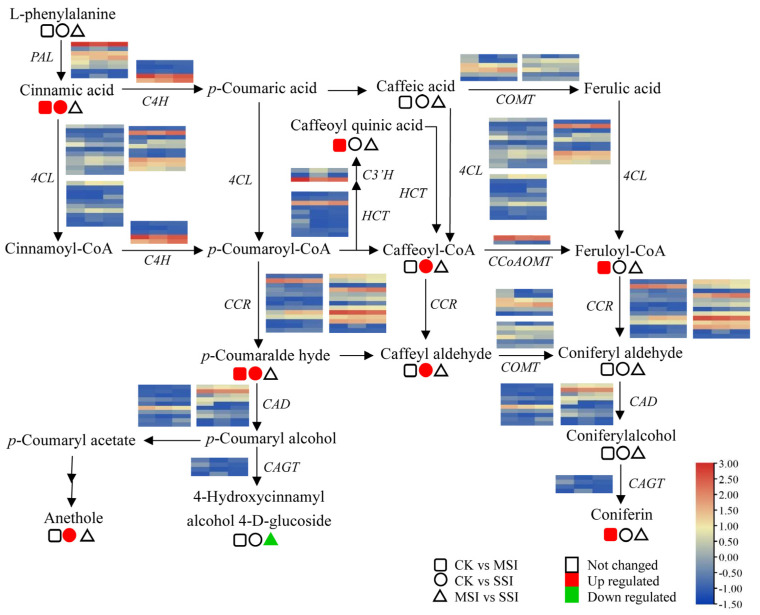
Changes of DEGs and DAMs in metabolic pathway of phenylpropanoid biosynthesis of Chinese olive under sunburn stress. The pathway of phenylpropanoid biosynthesis referring to the KEGG database (https://www.kegg.jp/kegg-bin/show_pathway?map00940) (accessed on 28 August 2023) [20]. Orange and blue colors indicate high and low expression levels of DEGs, while red and green colors represent the up- and down-accumulation of DAMs, respectively. No color indicates no significant change. Squares, circles, and triangles represent the comparisons of CK vs. MSI, CK vs. SSI, and MSI vs. SSI, respectively.

**Figure 5 plants-13-02369-f005:**
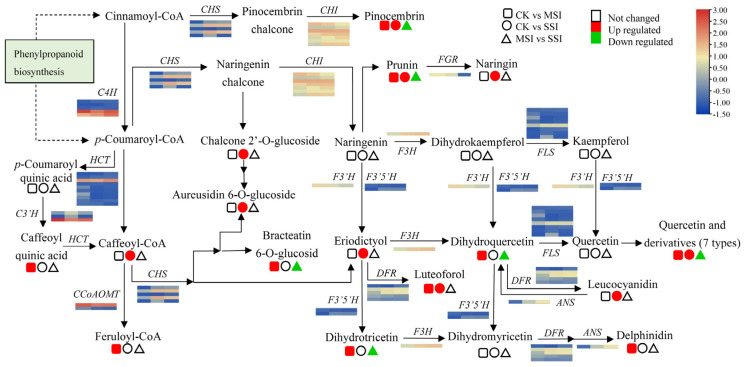
Changes in DEGs and DAMs in the metabolic pathway of flavonoid biosynthesis of Chinese olive under sunburn stress. The pathway of flavonoid biosynthesis referring to the KEGG database (https://www.kegg.jp/kegg-bin/show_pathway?map00941) (accessed on 28 August 2023) [20]. Orange and blue colors indicate high and low expression levels of DEGs, while red and green colors represent the up- and down-accumulation of DAMs, respectively. No color indicates no significant change. Squares, circles, and triangles represent the comparisons of CK vs. MSI, CK vs. SSI, and MSI vs. SSI, respectively.

**Figure 6 plants-13-02369-f006:**
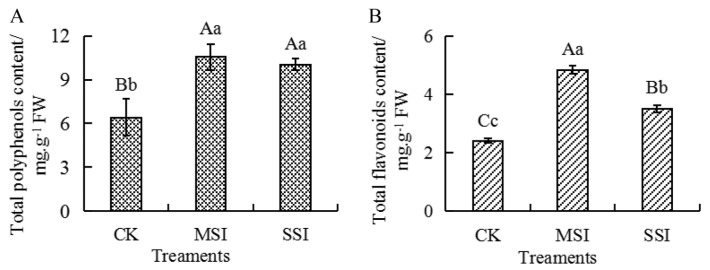
Changes in the content of total polyphenols (**A**) and total flavonoids (**B**) of Chinese olive fruits after sunburn stress. Upper- and lowercase letters on the curves represent significant differences at *p* value < 0.01 and <0.05 levels.

**Figure 7 plants-13-02369-f007:**
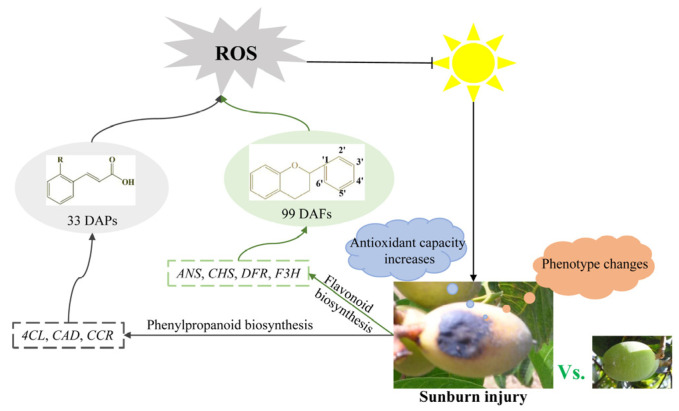
Potential regulation network of polyphenols and flavonoids in response to sunburn stress in Chinese olive.

## Data Availability

The data supporting reported results can be found in Appendix A. The raw sequence data reported in this paper have been deposited in the Genome Sequence Archive [53] in the National Genomics Data Center, China National Center for Bioinformation [54] (GSA: CRA012427) that are publicly accessible at https://ngdc.cncb.ac.cn/gsa (accessed on 29 August 2023).

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
