# Peer review of "Transcriptomic and Metabolomic Analysis Reveals the Potential Roles of Polyphenols and Flavonoids in Response to Sunburn Stress in Chinese Olive (*Canarium album*)"

_plants, 2024, doi:10.3390/plants13172369_

Round 1

Reviewer 1 Report

Comments and Suggestions for Authors

This manuscript considers the potential roles of polyphenols and flavonoids in Chinese olive in response to sunburn stress. Phenotypical changes, antioxidant capacity and antioxidative enzyme activities were evaluated.

Transcriptomic and metabolomic results were properly integrated and harmonized throught the text. 

While in this original study the analyses of polyphenolic metabolited and genes revieled 33 DAPs and 15 DEGs, a total of 99 DAPs and 8 DEGs were also determined from flavonoid biosynthesis pathway. 

Significant level of increases of total polyphenols and flavonoids were observed in the samples expossed to mild and severe sunburn injuries, compared to control samples. Authors suggested that the sunburn stress probably activates the transcription of structural genes involved in polyphenol and flavonoid biosynthesis, including 4CL, CCR, CAD, CHS, F3H, DFR, and ANS. This reviewer believes, it will be interesting for the reader to have some information about these genes. For instance;

- Are there any genes related to cell-wall structure (pectin, hemicellulose and/or cellulose)? Torres et al. Horticulture Research (2020) 7:22 determined photooxidative stress activates a complex multigenic response integrating the phenylpropanoid pathway and ethylene, leading to lignin accumulation in apple (Malus domestica Borkh.) fruit. Any relevence to these results?

- Authors have combined DEGs and DAMs in phenylpropanoid and flavonoid biosynthesis pathways (Figs 4 and 5) in a demonstrative way. However, this reviewer finds difficult to understand why mild sunburn exposure increased significantly more both the total polyphenols and total flavonoids than the severe exposure as compared to the control samples (no sunburn) (Fig 6). This should be discussed. Literature suggests early and delayed long-term transcriptional changes and short-term transient responses upon exposure to some stresses (María de la O Leyva-Pérez et al., DNA Research, 2015, 22(1), 1). While exposure gets more severe first level responses may not be adequate and plant may activate second level responses. Any relevant literature support/explanation or speculation related to the pathways involved may be helpful. 

Comments on the Quality of English Language

Minor typing errors detected;

-Line 32: Typos 'probably'

-Line 58: Space after full stop.

-Line 129: Rewrite last part.

-Line 183: Space after full stop.

-Line 218: Space after q.

-Line 432: Space after comma.

-Line 555: Consistency in tense use. Use past tense.

Reviewer 2 Report

Comments and Suggestions for Authors

Long et al. in their manuscript tried explain a potential role of polyphenols especially flavonoids in response to sunburn stress factor. The object of their research was Chinese olive fruit. The researches have analyzed this topic on transcriptomic and metabolomic levels. The authors cited 52 actual references (the oldest published in 1996).  The overall merit is good but I have several questions to authors. 

1. The methods section is clearly described, but I'm wondering about the choice of research material. The Chines olive in research came from the same experimental base and the same temperaure influencedfor this plants. Can we be sure that sunburn injury depends only from intensity of solar radiation? Maybe the different genotypes of these plants should be taken into account? Please answer.

2. The 4.2 section should be called methods (not materials).

3. The reults were clearly presented but in my opinion activity of antioxidant related enzymes and expression of coding theirs genes sholud be presented on one graph. 

Reviewer 3 Report

Comments and Suggestions for Authors

Long et al studied fruits of Chinese olive to explore the differentially expressed genes (DEGs) and differentially accumulated metabolites (DAMs) caused by different stress degrees including serious sunburn injury (SSI), mild sunburn injury (MSI), and ordinary (control check, CK) samples. The authors reported a considerable numbers of altered levels of polyphenols and flavonoids that were related to stress. It is nice study, but it could be improved further if the authors consider below listed comments.

1     Keywords and title have very similar wording please try to change keywords.

2      İntroduction seems pretty long it could be reduced.

3      Results/Discussion: Please do not start your sentence using numeric such as 3259 DEGS… 

4      Materials and Methods: Please revise this sentence “Five trees (numbered Ca1–Ca5)  that had been grafted for more than 9 years,..” do you mean 9 years old grafted plant?
